## Research Article

copper; membrane transporter; molecular dynamics; QM/MM; free energy

**Author for correspondence:**
*Alessandra Magistrato,
E-mail: alessandra.magistrato@cnr.it

# The conformational plasticity of the selectivity filter methionines controls the in-cell Cu(I) uptake through the CTR1 transporter

Pavel Janoš[1] , Jana Aupič[1] , Sharon Ruthstein[2] and Alessandra Magistrato[1]*

[1]Consiglio Nazionale delle ricerche/National Research Council (CNR) -IOM c/o International School for Advanced Studies (SISSA/ISAS), via Bonomea 265, 34136 Trieste, Italy and [2]Department of Chemistry, Faculty of Exact Sciences and the Institute for Nanotechnology and Advanced Materials (BINA), Bar-Ilan University, 5290002 Ramat-Gan, Israel

## Abstract

Copper is a trace element vital to many cellular functions. Yet its abnormal levels are toxic to cells, provoking a variety of severe diseases. The high affinity copper transporter 1 (CTR1), being the main in-cell copper [Cu(I)] entry route, tightly regulates its cellular uptake via a still elusive mechanism. Here, all-atoms simulations unlock the molecular terms of Cu(I) transport in eukaryotes disclosing that the two methionine (Met) triads, forming the selectivity filter, play an unprecedented dual role both enabling selective Cu(I) transport and regulating its uptake rate thanks to an intimate coupling between the conformational plasticity of their bulky side chains and the number of bound Cu(I) ions. Namely, the Met residues act as a gate reducing the Cu(I) import rate when two ions simultaneously bind to CTR1. This may represent an elegant autoregulatory mechanism through which CTR1 protects the cells from excessively high, and hence toxic, in-cell Cu(I) levels. Overall, our outcomes resolve fundamental questions in CTR1 biology and open new windows of opportunity to tackle diseases associated with an imbalanced copper uptake.

## Introduction

Copper is an essential metal for cellular growth and development. Copper acts as a signalling agent, promotes electron or oxygen transport and is a critical cofactor of copper enzymes involved in a wide variety of biochemical processes, including aerobic respiration, superoxide detoxification, hormone and neuropeptide biogenesis and connective tissue maturation (Magistrato *et al.,* 2019). As such, an imbalanced copper [Cu(I)] cellular uptake leads to metabolic abnormalities, anaemia, neurological, cardiac, connective tissue, immune disorders and Menkes' disease (Ahuja *et al.,* 2015). Conversely, Cu(I) overload is linked to Wilson's and neurodegenerative diseases (Prion's, Parkinson's and Alzheimer's) and is overwhelmingly associated to cancer onset and progression (Furukawa *et al.,* 2008; Denoyer *et al.,* 2015).

The transmembrane (TM) copper transporter 1 (CTR1) is an integral membrane protein mediating the selective Cu(I) uptake in all eukaryotes. CTR1, being the only known mammalian in-cell Cu(I) importer, plays a key regulatory role, tightly controlling the concentration of this trace metal to ensure proper cellular metabolism (Clifford *et al.,* 2016). Impaired CTR1-mediated Cu(I) transport causes delayed embryotic development, affects dietary Cu(I) acquisition and alters cardiac and liver functions (Ren *et al.,* 2019). As such, CTR1 is an appealing pharmacological target to tackle diseases linked to copper imbalance (Spinello *et al.,* 2021). Owing to its utmost importance, CTR1 has been attracting the interest of bioinorganic chemists for decades, but mechanistic advances were hindered by the low resolution of the available structural data (De Feo *et al.,* 2009; Schushan *et al.,* 2010). Only the recent crystallographic structure from *Salmo salar* (Ren *et al.,* 2019) supplied a blueprint to unlock the mechanistic intricacies of the CTR1-mediated Cu(I) transport.

Human CTR1 is a 190-amino acids-long homotrimer composed of three parts: (i) The N-terminal domain is responsible for catching Cu(II) from the extracellular space, its subsequent reduction to Cu(I) and transfer to the TM part; (ii) the TM part, which shuttles the Cu(I) ions across the membrane and (iii) the intracellular C-term part, which delivers Cu(I) to the intracellular chaperones. While the N-term and the C-term are disordered and not (or only partially) resolved in the X-ray structure (Ren *et al.,* 2019), the TM part is composed of three monomers, each arranged in a three-helix bundle fold and bearing two methionine (Met) residues (M150 and M154, in human) in a MX3M motif. Altogether, the three MX3M motifs compose two sets of Met triads, one extracellularly and one intracellularly oriented, which form the CTR1 selectivity filter and can host up to two Cu(I) ions. Mutational studies showed that the highly conserved MX3M motif plays a fundamental role in the Cu(I) transport, since M150L

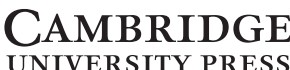

and M154L substitutions in human CTR1 abolish copper influx (Puig *et al.,* 2002). The role of these Mets is nevertheless ambiguous. On the one hand, they most likely ensure the exclusive uptake of Cu(I) ions due to the copper's chemical affinity for S-containing ligands; on the other hand, their bulky side chains can form a hydrophobic seal as observed in other channels (Ren *et al.,* 2019).

Here, by preforming enhanced sampling classical and quantum mechanical/molecular mechanical (QM/MM) molecular dynamics (MD) simulations, we unlock the molecular terms of the CTR1-mediated Cu(I) transport mechanism, disclosing an intriguing dual role of the selectivity filter Mets, which, owing to the conformational plasticity of their bulky side chains, strictly control the number and the rate of Cu(I) ions flowing through CTR1.

## Methods

The structure of CTR1, comprising the TM portion and part of the C-term (residues 41-186), was taken from the crystallographic structure Protein Data Bank (PDB) entry: 6M98 from *S. salar* (Fig. 1*a*; Ren *et al.,* 2019). The chaperone apocytochrome b$_{562}$RIL (BRIL), present in the crystal structure, was replaced by the native loop modelled with the MODELLER program (Webb and Sali, 2016). A total of three models were built: one with only one Cu(I) ion bound to the most extracellularly exposed Met150 (M154 in hCTR1) triad (Site 1), one with a Cu(I) bound only to the intracellularly exposed Met146 (M150 in hCTR1) triad (Site 2) and one with two Cu(I) ions bound to Site 1 and Site 2 as observed in the crystal structure (Fig. 1*b*). CHARMM-GUI (Jo *et al.,* 2008;

Wu *et al.,* 2014) was used to insert the CTR1 protein into a lipid bilayer and create the simulation box. The protein was described using Amber FF14SB force field (FF; Maier *et al.,* 2015), the 1-palmitoyl-2-oleoyl phosphatidylcholine (POPC) bilayer using lipids17 FF (Dickson *et al.,* 2014) and water using the TIP3P model (Jorgensen *et al.,* 1983). Joung and Cheatham (2008) parameters were used for Na$^+$ and Cl$^-$ ions. Nonbonded Cu FF parameters were taken from Merz and coworkers (Op't Hol and Merz, 2007), and the Cu(I) ions together with the Met triads were restrained using a matrix of distance restraints to maintain the coordination geometry as close as possible to the initial geometry during the 100-ns-long classical MD simulations performed to relax the system before switching to QM/MM MD simulations.

The QM/MM MD simulations were performed using CP2K version 6.1 (Hutter *et al.,* 2014). The QM zone comprised all six Met residues (cut at the C$\alpha$–C$\beta$ bond) forming the selectivity filter and one or two Cu(I) ions along with nearby water molecules. The QM zone was described using Becke-Lee-Yang-Parr (BLYP) functional (Becke, 1988; Lee *et al.,* 1988; Miehlich *et al.,* 1989) with dual Gaussian-type/plane waves basis set as in previous simulations (Pavlin *et al.,* 2019; Qasem *et al.,* 2019). Specifically, a double-$\zeta$ molecularly optimized (MOLOPT) basis set (VandeVondele and Hutter, 2007) was used with an auxiliary plane-wave (PW) basis set using the Goedecker–Teter–Hutter pseudopotentials (Goedecker *et al.,* 1996) and a plane wave cut-off of 320 Ry. DFT-D3 dispersion correction was applied (Grimme *et al.,* 2010). The metadynamics (MTD) simulations were exploited to study the mechanism of Cu(I) translocation through the CTR1 selectivity filter using the MTD implementation of CP2K. This was done by using two

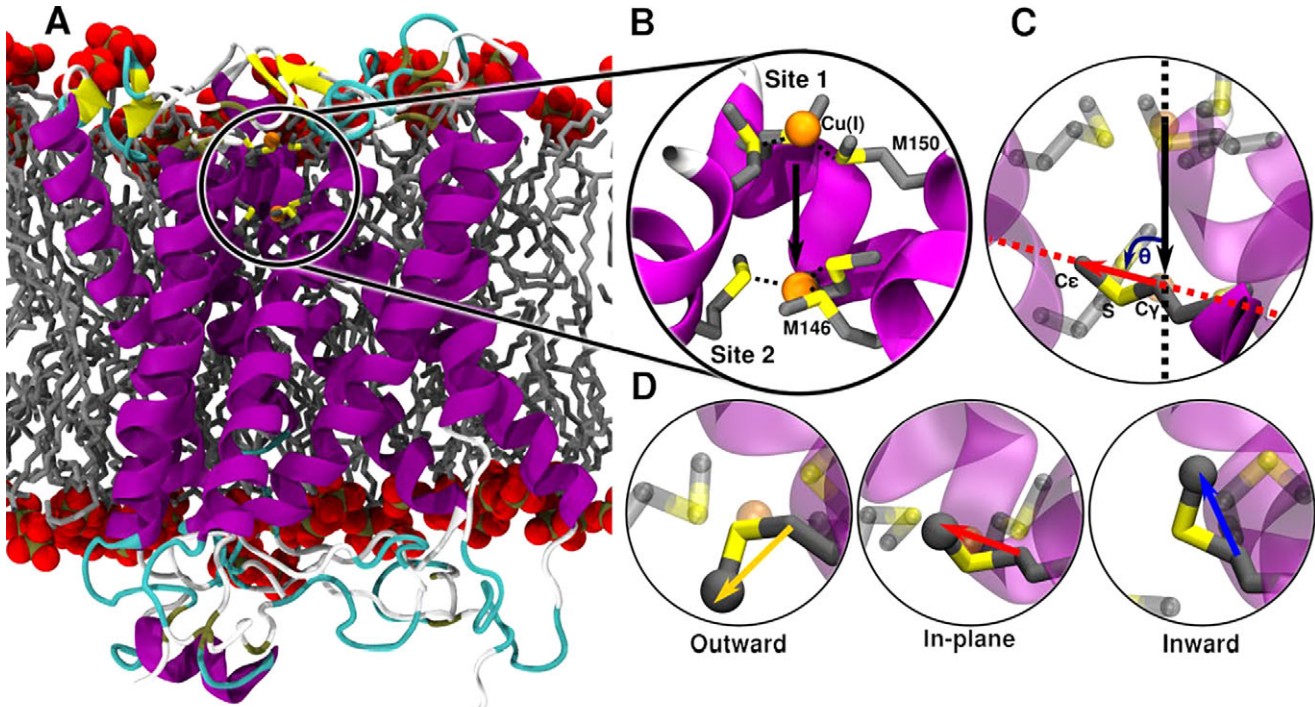

**Figure 1.** (*a*) Structure of CTR1 transporter from *Salmo salar* built on the X-ray structure (PDB code 6M98). CTR1 is shown in new cartoon representation, with magenta *α*-helices, yellow *β*-sheets and cyan loops. The membrane is represented as grey liquorice, with the phosphate groups displayed in van der Waals spheres. (*b*) Close-up of the Met triads with both Site 1 [top Met150 (M154 in hCTR1) triad] and Site 2 [bottom Met146 (M150 in hCTR1) triad] occupied with Cu(I) ions. Met residues are shown in liquorice with S and C atoms in yellow and grey, respectively. Hydrogen atoms are omitted for clarity. Cu(I) is depicted as an orange van der Waals sphere. Cu(I) coordination sphere is highlighted with dashed black lines. The black arrow indicates the vector of the selectivity filter defined by the centres of mass of the Site 1 and Site 2 Mets backbone atoms. (*c*) Definition of the θ angle used to classify the Met conformations: the black line indicates the vector of the selectivity filter and the red line indicates the Met C$\gamma$–C$\varepsilon$ vector. (*d*) The three possible conformations of the selectivity filter Site 2 Met146 with the C$\varepsilon$ atom highlighted as grey sphere: outward, in-plane and inward states highlighted with yellow, red and blue arrows, respectively.

collective variables (CVs): CV1 – distance of the Cu(I) to the plane of the Site 1 Met-150 triad, defined by the Met150 $C\alpha$ atoms; and CV2 – defined as the coordination number (CN) of the Cu(I) ion with respect to sulphur (S) atoms of the Site 2 Met146 triad, similarly to a previous study (Janoš and Magistrato, 2021). The dissociation of the Cu(I) ion from the Site 2 Met146 triad was simulated with the MTD implementation in Plumed 2.7.0 (Tribello *et al.,* 2014) using again two CVs: CV1 – accounting for the *z*-projection of the distance of the Cu(I) to the centre of the CTR1 selectivity filter defined using $C\alpha$ atoms of all Site 1 and Site 2 Mets; and CV2 – defined as CN of the Cu(I) with respect to S atoms of the Site 2 Met146 triad. The final dissociation step from the partially dissociated state, in which Cu(I) is still coordinated to one Site 2 Met, was explored with a separate set of MTD simulations using only one CV (CV1) – the distance of the Cu(I) ion to the S atom of the last coordinated Site 2 Met.

The classical MTD simulation of Met flipping was performed using specific bonded Cu(I) and Met FF parameters that we derived by using the Metal Center Protein Builder (Li and Merz, 2016) on the QM/MM equilibrated CTR1 structure.

For analysis purposes, the three possible conformational states of Mets were defined by the angle θ between two vectors: the vector lying between geometric centres of Site 1 and Site 2 Met backbone atoms (Fig. 1c) representing the axis of the selectivity filter; and the second vector defined by Cγ–Cε Met atoms (Fig. 1c). The Site 1 conformations are defined with θ angle ranges: 0–40° for the Outward (O), 50–90° for In-Plane (IP) and 100–160° for Inward (I) conformations of Met150 residues. The Site 2 conformations are defined with θ angle ranges: 100–160° for the Outward (O), 50–90° for IP and 0–40° for Inward (I) conformations of the Met146 residues.

A more detailed description of the systems setup, MD and MTD simulations are provided in the Supplementary Material.

## Results

### Binding mode of Cu(I) ions and conformational response of the selectivity filter

The crystal structure of CTR1 shows two Cu(I) ions bound to the selectivity filter, one per each Met triad, with the Cu(I)–S distances in the upper (most-extracellularly exposed) Met150 triad (hereafter referred to as Site 1) and in the bottom (most-cytosol exposed) Met146 triad (hereafter referred to as Site 2) being 2.16 and 3.22 Å, respectively. The large distances between the Cu(I) ion and the sulphur atoms at Site 2 suggest that the metal ion is not (or is only weakly) bound to this site. The thioether group of the Met residues forming the triads supply a trigonal planar coordination geometry, presumably ensuring the selective uptake of Cu(I) over Cu(II) ions. Both Cu(I) ions are placed below the plane of the Met triad to which they bind, laying at a distance of 7.22 Å from each other.

In order to refine the crystal structure and establish the number and the coordination geometry of metal ions, which can simultaneously bind to the transporter, we performed a 100-ns-long classical MD equilibration of *S. salar* CTR1 (PDB 6M98), described by Amber FF14SB (Maier *et al.,* 2015; FF), embedded in a lipid membrane mimic, described by Lipid17 FF (Dickson *et al.,* 2014), with two bound Cu(I) ions (Fig. 1a), which was then followed by 10-ps QM/MM MD (Janoš *et al.,* 2021) using CP2K (Hutter *et al.,* 2014) with DFT-BLYP functional. The resulting average Cu(I)–S coordination distance for both triads is 2.32 ± 0.16 Å (Supplementary Fig. 1), whereas the Cu(I)–Cu(I) distance

decreases to a mean value of 6.18 ± 0.42 Å (Fig. 1b and Supplementary Fig. 2). In order to assess whether the two Cu(I) ions may be spaced out by water molecules during their permeation, as in ions channels (Napolitano *et al.,* 2015), we placed a water molecule between them. This invariably triggered a distortion of the selectivity filter (data not shown), leading us to rule out this possibility. As such, this set of simulations suggests that CTR1 may simultaneously bind two Cu(I) ions in the selectivity filter, even though the resulting coordination geometry is slightly different from that captured in the crystal structure with reduced Cu(I)–Cu(I) and Cu(I)–S coordination distances in the bottom triad (Supplementary Fig. 3). We hypothesize that crystallographic structure corresponds to a state with a partially dissociated second Cu(I) ion (see the 'Cu (I) dissociation from Site 2 in the presence of Cu(I) ion bound to Site 1' section). Furthermore, we investigated the binding of only one Cu(I) ion to either the Site 1 or Site 2 Met triad. In this case, the Site 2 or Site 1 bound ion, respectively, was removed from the previous model, which was then subjected to additional QM/MM MD simulations. The coordination geometries obtained in the presence of a single Cu(I) ion were similar to those of the two Cu(I) ions bound model, but a remarkable difference in the conformational plasticity of the selectivity filter Mets emerged. First, the root mean square deviation (RMSD) and root means square fluctuation (RMSF) analyses of these residues pinpointed striking asymmetries between the two Met triads (Supplementary Figs S4 and S5) in the distinct models. Namely, while the overall flexibility of Site 1 Met triad increases when CTR1 hosts two Cu(I) ions in the selectivity filter as compared to the Site 1-only bound model, the flexibility of one of the Site 2 Mets layers drastically decreases in the presence of two Cu(I) ions with respect to the Site 2-only bound model (Supplementary Fig. 5).

A detailed inspection of the conformational behaviour of the selectivity filter Mets revealed that these residues can assume three different conformations (Fig. 1d): (i) an Outward (O) state, where the S-methyl bond is parallel to the *z*-axis of the selectivity filter and the methyl group points away from the selectivity filter centre (i.e. it points towards the extracellular side in Site 1 and towards intracellular CTR1 vestibule in Site 2); (ii) an In-Plane (IP) state, in which the S-methyl bond is perpendicular to the *z*-axis of the filter and the methyl group points away from the selectivity filter centre and (iii) an Inward (I) state with the S-methyl bond parallel to the *z*-axis of the selectivity filter and with the methyl group pointing inside the selectivity filter.

The analysis of Mets conformational behaviour (Fig. 2) disclosed clear differences among the distinct models investigated. In the apo model, the Mets residues access all conformations, even though a coupling exists between the Site 1 and Site 2 Met triads with only one Site 1 Met at a time filling the selectivity filter by assuming the I-conformation. When only Site 1 is occupied, two coordinating Mets assume IP-conformation and one lies in between the IP-state and the O-state. Conversely, the Mets of the empty Site 2 are more flexible with one laying in the IP-state, one between the IP-state and the I-state and one toggling around the IP-conformation reaching at times the I-state or O-state (Fig. 2). Interestingly, when only Site 2 is occupied, two Mets coordinating the Cu(I) ion retain the IP-conformation, with one Met still toggling around the IP-state, and visiting more frequently the O-conformation. Conversely, two Mets of the empty Site 1 assume the IP-conformation and the O-state (Fig. 2). The latter Met most likely hinders the access of additional Cu(I) ions to the selectivity filter from the extracellular matrix. Stunningly, when both sites bind a Cu(I) ion, all Site 1 Mets rigidly retain the IP-conformation,

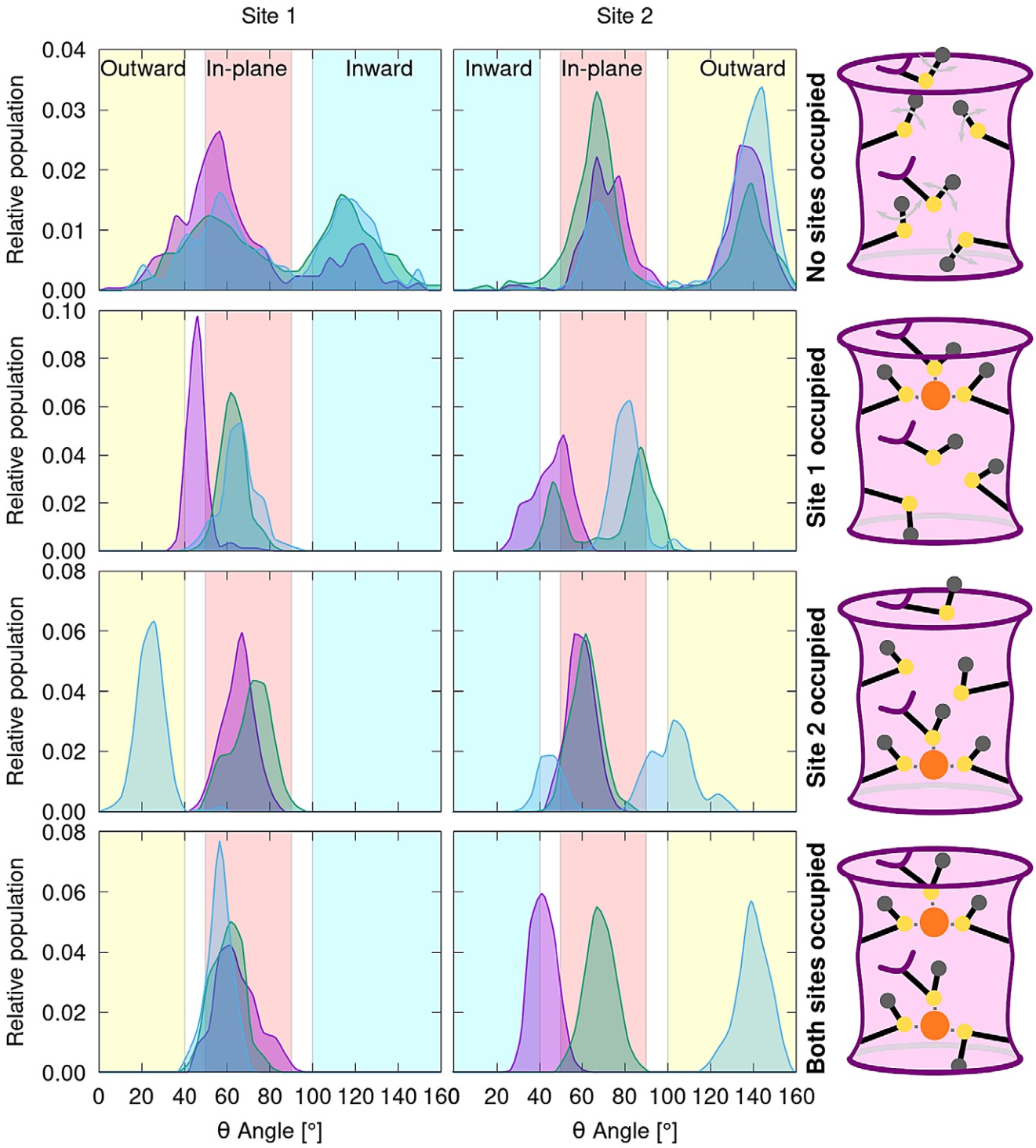

**Figure 2.** Distribution of the θ angle (°) (defined in Fig. 1*c*) as obtained from QM/MM and classical [for apo CTR1, containing no bound Cu(I) ions] MD trajectories. The distribution is shown for apo CTR1 model (top panel), CTR1 with one Cu(I) ion bound to Site 1 (second panel from top); to Site 2 (third panel from top), and with two Cu(I) ions bound to both sites (bottom panel). Left and right columns show the angle distribution of the Site 1 and Site 2 Met triads, respectively. The angle distributions of the three Met1-3 residues are shown with different colours (magenta, green and blue, respectively). Areas of the histogram corresponding to the inward, in-plane and outward conformations are highlighted in cyan, red and yellow, respectively. Sketches of the conformational behaviour of the different models are shown on the right. The CTR1 selectivity filter is schematically depicted as a pink cylinder with violet walls. The Mets residues are schematically represented in black lines with sulphur, Cu(I) and Cε atoms highlighted in yellow, orange and grey circles, respectively. Cu(I) coordination sphere is shown with dashed grey lines. In apo CTR1, grey arrows indicate the Met conformational motion.

whereas one of the Site 2 Mets lays between the I- and IP-conformation, one in the IP-conformation and the last one remains rigidly trapped in the O-conformation. This reveals that the conformations assumed by the two Met triads are not only coupled, but

also intimately connected to the number of bound Cu(I) ions. As such, their Cu(I)-induced conformational selection may play a key role in the CTR1 transport mechanism. To further elucidate this intriguing trait, in the following, we investigated the Cu(I) transport

mechanism through CTR1 in the presence of one or two Cu(I) ions bound to the selectivity filter.

## Cu(I) translocation from the Site 1 to the Site 2 Met triad

We initially assessed the early events of CTR1-mediated transport by monitoring the translocation of a Cu(I) ion from Site 1 to Site 2. CTR1 model initially hosted Cu(I) in Site 1, and the CTR1 import mechanism was investigated by performing QM/MM MTD simulations using two CVs. The first (CV1) was defined as the distance

of the Cu(I) ion from the plane of the Site 1 Met triad, and the second (CV2) as the CN of the Cu(I) ion with respect to the S-atoms of the Site 2 Met triad (Fig. 3a,b). These MTD simulations unveiled that the Cu(I) ion, by surmounting a Helmholtz free energy barrier ($\Delta F^{\ddagger}$) of $5.8 \pm 2.8$ kcal mol$^{-1}$, heads from Minimum 0 (M0), where it is coordinated by the three Site 1 Mets, all in the IP-state, to M1, lying at a free energy ($\Delta F$) of $-4.9 \pm 2.8$ kcal mol$^{-1}$, where Cu(I) is bound by two Site 1 Mets, still in the IP-state, and one Site 2 Met also in the IP-state. Next, Cu(I) moves to M2 ($\Delta F^{\ddagger} = 6.1 \pm 1.1$ kcal mol$^{-1}$ and $\Delta F = -3.9 \pm 0.2$ kcal mol$^{-1}$),

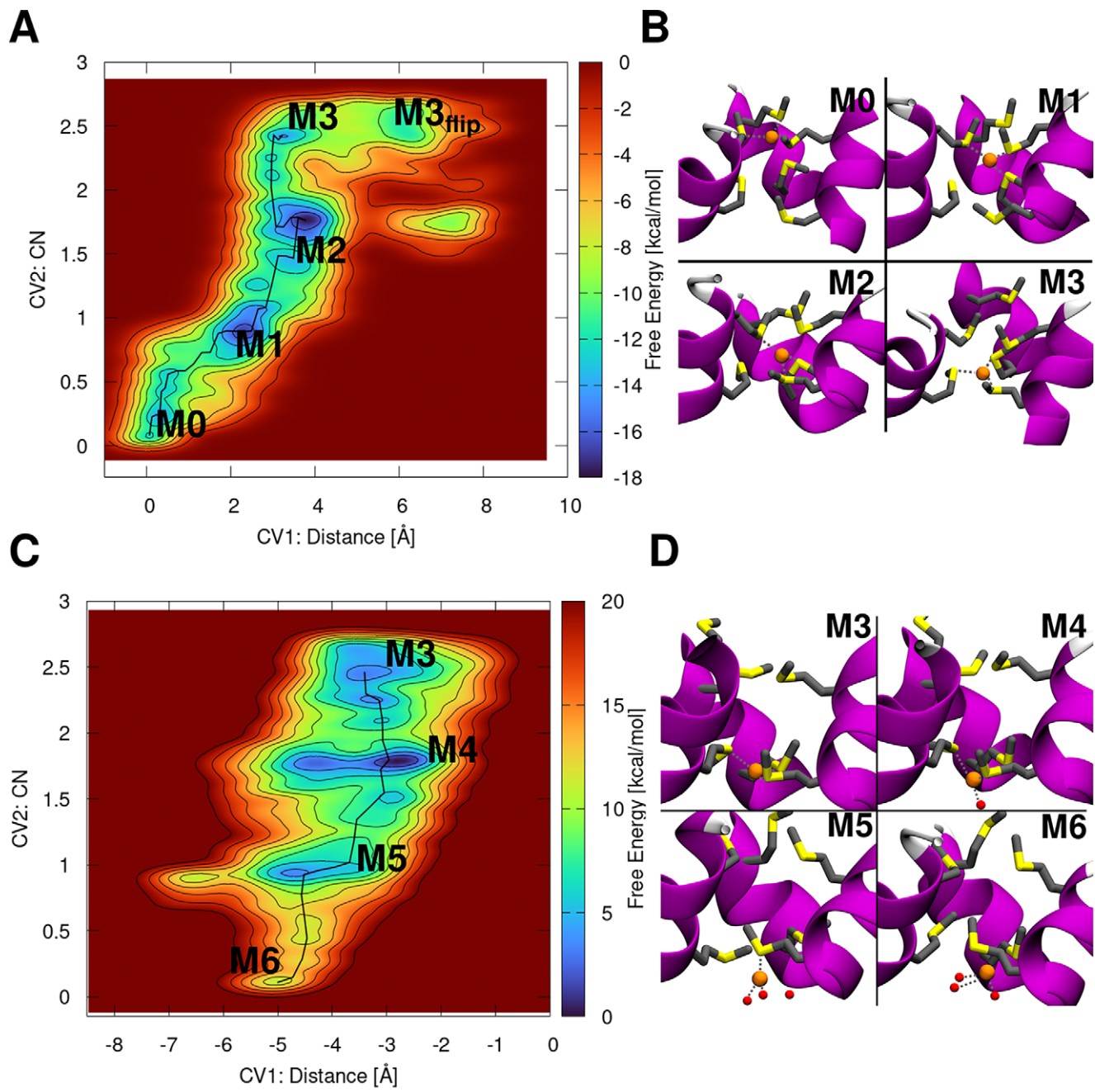

**Figure 3.** (a) Free energy surface (FES; kcal mol$^{-1}$) of the Cu(I) translocation from Site 1 to Site 2 plotted as function of Cu(I) distance from the Site 1 Met150 triad (CV1) and coordination number (CN) of Cu(I) with respect to the bottom Site 2 Met146 triad (CV2). (b) Close-ups of the minima visited during the Site 1 to Site 2 translocation. (c) FES of the Cu(I) dissociation from Site 2 in absence of Cu(I) in site 1 plotted as a function of the *z*-projection of the Cu(I) distance (Å) from the centre of the selectivity filter defined by C$\alpha$ atoms of the Site 1 and Site 2 Mets [Collective Variable 1 (CV1)]; and CN of Cu(I) with respect to the Site 2 Met triad (CV2). (d) Close-up views of Cu(I) dissociation states from Site 2. Both FESs are shown from blue to red with isosurface lines drawn every 2.0 kcal mol$^{-1}$. CTR1 is shown as magenta new cartoons, Met triads as liquorice and Cu(I) ion and water molecules as orange and red spheres, respectively. Hydrogens are omitted for clarity. Cu(I) coordination is highlighted with dashed grey lines.

where it is bound by one Site 1 Met and two Site 2 Mets, all in the IP-state. Interestingly, the Site 1 Met1, still coordinating the metal, moves down within the selectivity filter to follow Cu(I), while maintaining the IP-conformation (Supplementary Fig. 6C,E). The Cu(I) ion ultimately reaches M3 ($\Delta F^{\ddagger}$ = 4.3 ± 4.0 kcal mol$^{-1}$), where it is coordinated by the three Site 2 Mets, all in the IP-state (Supplementary Fig. 6C,D).

The additional minimum observed in the free energy surface (M3$_{flip}$; Supplementary Fig. 6) is associated with the conformational switch of one Site 2 Met from IP- to O-conformation, which occurs after Cu(I) is fully transferred to Site 2 (Supplementary Fig. 6D,F). A $\Delta F^{\ddagger}$ = 5.9 ± 1.0 kcal mol$^{-1}$ separates M3 from M3$_{flip}$. Even if in M3$_{flip}$ the Cu(I) ion has moved further down towards the cytosolic exit as compared to M3, suggesting that M3$_{flip}$ might be an early intermediate along the Cu(I) dissociation pathway from Site 2 to the cytosol, the O-conformation of the Site 2-Met may prevent the access of water to the selectivity filter and thus the Cu(I) dissociation towards the cytosol-exposed cavity of CTR1. As such, in the following, we investigated the Cu(I) dissociation from the selectivity filter starting from M3, which is more thermodynamically stable than M3$_{flip}$, displays all Site 2 Mets in the IP-state and allows for the water molecules to easily access the Cu(I) coordination sphere (Supplementary Fig. 7), thus facilitating the Met/water exchange reaction and possibly the Cu(I) dissociation towards the cytosolic cavity.

### In-cell Cu(I) dissociation mechanism

From M3 state, two possible scenarios open: the Cu(I) ion can dissociate from the selectivity filter to the cytosol before or after the binding of a second Cu(I) ion to Site 1.

### Cu(I) dissociation from Site 2 in the absence a Cu(I) ion bound to Site 1

We initially assumed that a single Cu(I) ion at a time flows through CTR1 and explored the dissociation of Cu(I) from Site 2 in the absence of any Cu(I) ion bound to Site 1. To this aim, we performed QM/MM MTD simulations using as CVs the *z*-projection of the Cu(I) distance from the centre of CTR1 selectivity filter (CV1) and the CN of Cu(I) with respect to the Site 2 Met-triad sulphur atoms (CV2; Fig. 3*c,d*). This MTD simulation was started from the equilibrated structure of the M3 state. From M3, the Cu(I) ion in-cell release starts by overcoming a $\Delta F^{\ddagger}$ = 5.8 ± 2.6 kcal mol$^{-1}$ barrier to reach the M4 state ($\Delta F$ = −3.0 ± 2.7 kcal mol$^{-1}$), where Cu(I) acquires a water molecule in the coordination sphere, after losing one Site 2 Met. Next, Cu(I) reaches M5 ($\Delta F^{\ddagger}$ of 8.5 ± 2.7 kcal mol$^{-1}$ and $\Delta F$ = 3.0 ± 3.2 kcal mol$^{-1}$), where it is hydrated by two water molecules, while being still bound to a single Site 2 Met. Finally, the Cu(I) ion reaches M6, in which it is coordinated by three waters [$\Delta F^{\ddagger}$ = 12.5 ± 1.2 kcal mol$^{-1}$ (Supplementary Fig. 9)[1] and $\Delta F$ = 8.1 ± 3.0 kcal mol$^{-1}$]. The Site 2 Met1, still coordinating the Cu(I) ion in M5, is very flexible, toggling between IP-, I- and O-conformations during the dissociation process (Supplementary Fig. 8), whereas the other two Met residues remain in the IP-state (M4–M6 states). This highlights how the Cu(I)-dependent conformational remodelling of the selectivity filter Mets takes an active role during Cu(I) transport.

---

[1]The free energy barrier obtained from the original MTD simulations based on two collective variables is of 9.8 ± 3.5 kcal mol$^{-1}$.

### Cu(I) dissociation from Site 2 in the presence of Cu(I) ion bound to Site 1

We finally explored the dissociation of Cu(I) from Site 2, when also Site 1 binds a Cu(I) ion, by using the same set of CVs as detailed above (Fig. 4). In the initial M$_{2Cu}$0 state, the two Cu(I) ions force one Site 2 Met to assume a conformation in-between the IP- and I-states, whereas the remaining two Mets adopt the IP- and O-conformations (referred to as Met1, Met2 and Met3, respectively). Next, the metal dissociates from Site 2 by initially losing the coordination with the Site 2 Met1 in the I-state (M$_{2Cu}$1; $\Delta F^{\ddagger}$ = 10.8 ± 2.7 and $\Delta F$ = 8.9 ± 2.4 kcal mol$^{-1}$). This is followed by a water exchange reaction with the Site 2 Met3 in the O-state reaching the M$_{2Cu}$2 intermediate ($\Delta F^{\ddagger}$ = 18.9 ± 3.1 kcal mol$^{-1}$ and $\Delta F$ = 17.8 ± 2.9 kcal mol$^{-1}$; Fig. 4). Noticeably, in M$_{2Cu}$2, the Cu(I)–Cu(I) distance as well as the distances between Cu(I) and the Site 2 Met residues resemble those captured in the X-ray structure (Ren *et al.,* 2019), raising the possibility that the experimental structure captured a state in which the Cu(I) ion in Site 2 is partially dissociated.

Stunningly, in this intermediate state, the Site 2 Met2, which is the last Met coordinating Cu(I), is rather flexible, approaching the O-state during the final dissociation step (Supplementary Fig. 10), whereas Site 2 Met3, which permanently adopts the O-conformation, occludes the water access to the selectivity filter, thus hampering the Met/water exchange reaction (Supplementary Fig. 7) and hindering the complete dissociation of Cu(I) from Site 2. To assess how the Site 2 hydration affects the energetics of the Cu(I) in-cell release, we performed additional MTD simulations using as CV the distance of Cu(I) to the Site 2 Met2. To this aim, we built two different models based on M$_{2Cu}$2 intermediate: one with the Site 2 Mets free to move (i.e. initially assuming one the O-state, and two the IP-state); and a second model in which all Site 2 Mets were restrained to the IP-conformation (Supplementary Fig. 12). The resulting $\Delta F^{\ddagger}$s to reach the fully dissociated M$_{2Cu}$3 state were of 32.2 ± 2.7 and 21.3 ± 3.3 kcal mol$^{-1}$ for the two models, respectively (Supplementary Fig. 12), confirming that the Met3 in the O-state hinders Cu(I) in-cell release. Nevertheless, the reduced conformational plasticity of the Mets, induced by restraining them in the IP-conformation, also contributes to slowing the Cu(I) in-cell release (increasing the $\Delta F^{\ddagger}$ to 21.3 ± 3.3 kcal mol$^{-1}$) as compared to the Cu(I) dissociation from the Site 2-only bound model where all Mets were in the IP-state without any constraints ($\Delta F^{\ddagger}$ = 12.5 ± 1.2 kcal mol$^{-1}$; Fig. 3*c*). Furthermore, we monitored the Gibbs free energy cost ($\Delta G^{\ddagger}$) of flipping a single Site 2 Met from the O- to IP-state, in the two Cu(I) ions bound model, by performing classical MTD simulation using the dihedral angles χ1, χ2 and χ3 as CVs (Supplementary Fig. 13). The resulting $\Delta G^{\ddagger}$ was 8.6 ± 1.9 kcal mol$^{-1}$, further indicating that the Met conformation flipping occurs at a non-negligible free energy cost.

These findings compellingly show that the Cu(I) binding to both Met triads reciprocally affects their conformational plasticity, with the Mets side chains acting as gate to limit the amount of in-cell Cu(I) import.

### Discussion

Our atomic-level dissection of the CTR1-mediated Cu(I) in-cell uptake mechanism discloses that during the early steps of Cu(I) translocation from Site 1 to Site 2, the metal ion visits two stable intermediates in which Cu(I) is concomitantly coordinated to both Site 1 and Site 2 Mets. The progression between these two

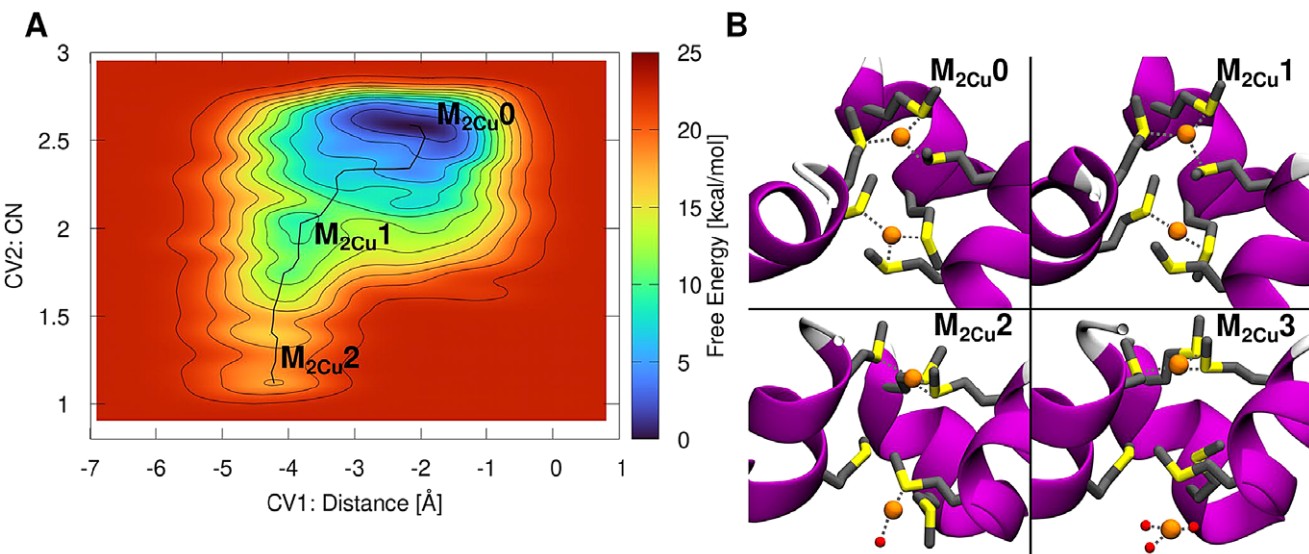

**Figure 4.** Mechanism of the Cu(I) dissociation from Site 2 in presence of Cu(I) in Site 1. (a) Free energy surface (FES; kcal mol$^{-1}$) plotted as function of the *z*-projection of the Cu(I) distance (Å) from the centre of the selectivity filter defined by C$\alpha$ atoms of the Site 1 and Site 2 Mets [Collective Variable 1 (CV1)]; and coordination number of Cu(I) with respect to the Site 2 Met-triad sulphur atoms (CV2). The FES is shown from blue to red with isosurface lines drawn every 2.0 kcal mol$^{-1}$. (b) Close-ups of the states visited during Cu(I) dissociation from Site 2 (state M$_{2Cu}$3 obtained from additional 1D MTD; Supplementary Fig. 12). CTR1 is shown as magenta new cartoon, Met triads as liquorice and Cu(I) ion and water molecules as orange and red spheres, respectively. Hydrogens are omitted for clarity. Cu(I) coordination is shown with grey dashed lines.

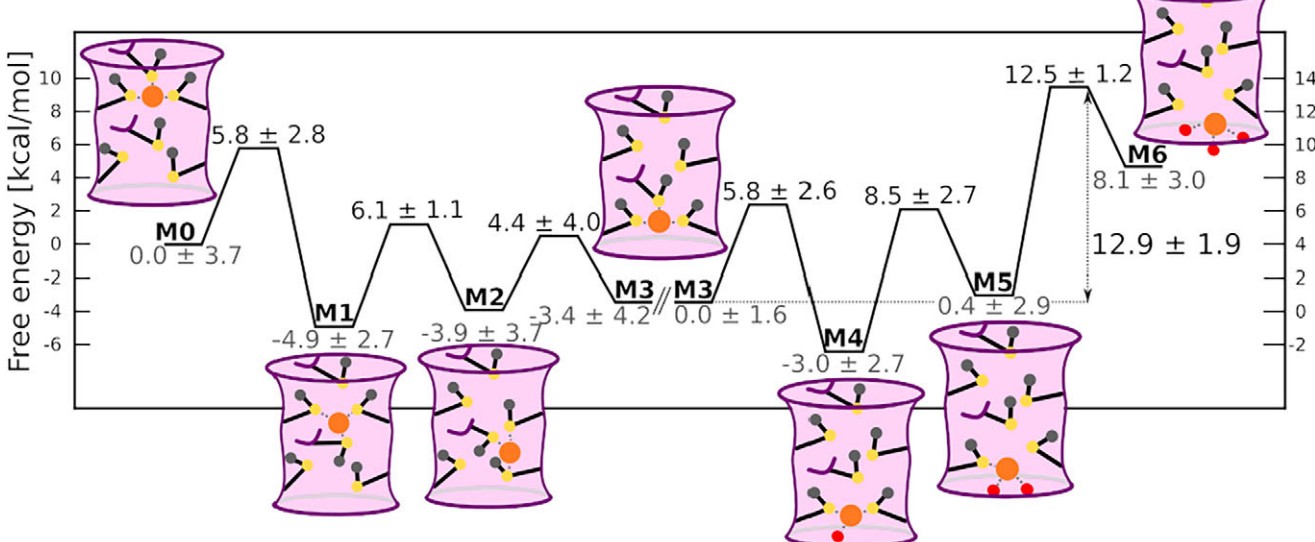

**Figure 5.** Schematic representation of the CTR1-mediated Cu(I) translocation processes. Free energy barriers of each step are reported (kcal mol$^{-1}$). The CTR1 selectivity filter is schematically depicted as a pink cylinder with violet walls. The Met residues are schematically represented in black lines with sulphur, Cu(I) and water oxygen atoms highlighted with yellow, orange and red circles, respectively. Cu(I) coordination sphere is shown with dashed grey lines. Parallel diagonal lines separate the results of the Cu(I) translocation within the selectivity filter from those of Cu(I) release to the cytosol. The left and right axes refer to the free energy cost for translocation and dissociation mechanism, respectively.

minima displays the highest $\Delta F^{\ddagger}$ (6.1 $\pm$ 1.1 kcal mol$^{-1}$) for Cu(I) movement within the selectivity filter (Fig. 5). Nevertheless, the rate-limiting step of the in-cell transport process corresponds to the Cu(I) dissociation from the selectivity filter towards the cytosol-exposed CTR1 vestibule. The calculated $\Delta F^{\ddagger} = 12.9 \pm 1.9$ kcal mol$^{-1}$ (Fig. 5) of this step is in good agreement with experimental turnover rate of 6.0–14.2 s$^{-1}$ at 37°C (Maryon *et al.,* 2013) corresponding to a $\Delta G^{\ddagger}$ of 13.0–13.5 kcal mol$^{-1}$.

Most importantly, our findings unprecedentedly unlock an intimate coupling between the Site 1 and Site 2 Met triad

occupancy and conformational plasticity as a salient CTR1 trait. Owing to this coupling, one Site 2 Met adopts an O-conformation when two Cu(I) ions are bound to CTR1. This prevents Cu(I) hydration [i.e. water access to the Cu(I) bound at the intracellular exit of the selectivity filter], significantly increasing the free energy barrier ($\Delta F^{\ddagger} = 32.2 \pm 2.7$ kcal mol$^{-1}$) for the Cu(I) in-cell release as compared to the same step in the Site 2-only bound model ($\Delta F^{\ddagger} = 12.5 \pm 1.2$ kcal mol$^{-1}$). Altogether, our outcomes disclose that CTR1 leverages the conformational coupling of the Met layers in the selectivity filter to selectively propel

Cu(I) in-cell uptake in a limited and healthy amount [i.e. enabling the translocation of a single Cu(I) ion at a time]. In this scenario, it is tempting to argue that when two Cu(I)-ions bind to CTR1, as it may potentially occur at high extracellular copper concentrations, the conformational coupling of the two selectivity Met-triads may be an elegant way of limiting the Cu(I) import rate. This may thus represent a sophisticated autoregulatory mechanism to protect cells from excessively high, and hence toxic, in-cell Cu(I) levels.

The importance of dynamic properties of coordinating residues, dictated by local interactions, for selective ion transport has been highlighted before, notably for the potassium channel KcsA (Noskov *et al.*, 2004). Similarly to CTR1, KcsA contains multiple metal binding sites; however, binding of multiple potassium ions is thought to ensure highly efficient ion conduction via so-called knock-on mechanism mediated by electrostatic repulsion between potassium ions (Köpfer *et al.*, 2014). Contrary, we find that binding of multiple Cu(I) ions in the selectivity filter of CTR1 decreases the copper influx rate, most likely reflecting the different roles potassium and copper ions play in the cellular milieu.

Met residues were shown to participate in ion channel gating previously, albeit as a hydrophobic barrier, precluding the ion passage indirectly (Neale *et al.*, 2015; Rao *et al.*, 2021). In CTR1 instead, a sophisticated and clever interplay between metal coordination and Met conformational plasticity emerges. Interestingly, Met-mediated Cu(I) transport is shared also by the bacterial CusA efflux pump relying on three sets of Met-dyad binding sites to rapidly export Cu(I). However, these Mets do not seem to play a regulatory role, most likely due to the bacterial need to rapidly expel toxic Cu(I) ions from the cells to guarantee their survival (Long *et al.*, 2010). This observation further supports the hypothesis that the third Met in each binding site may be necessary to achieve autoregulatory function.

## Conclusion

In this study, QM/MM MD-enhanced sampling simulations resolve the intricate mechanism underlying the Cu(I) import through CTR1, compellingly assigning a new dual functional role to the selectivity filter Met residues. These residues selectively bind Cu(I) ions thanks to the affinity of their sulphur atoms for copper and also act as gates to regulate Cu(I) import uptake in a controlled and healthy amount (Ren *et al.*, 2019). Namely, our outcomes show that the selectivity filter Mets act as gates in response to Cu(I) overload thanks to a sophisticated coupling between the amount of Cu(I) ions bound to CTR1 and the conformational response of the Met side chains. Indeed, the simultaneous binding of two Cu(I) ions to the selectivity filter activates the Met gate, reducing the Cu(I) ions flow rate to avoid their imbalanced, and hence toxic, accumulation in the cell. The detailed atomic-level picture of CTR1 import mechanism supplied here sets a conceptual basis to develop novel mechanism-based therapeutics tackling the variety of human diseases entwined to an inappropriate Cu uptake.

**Supplementary material.** Extended Methods description; Supplementary Figs S1–S13: Cu(I)-S and Cu(I)–Cu(I) distances, Met-triads structure, RMSD and RMSF of Site 1 and Site 2, Cu(I) translocation from Site 1 to Site 2, RDF of Cu(I)-O$_{water}$ distances, Cu(I) dissociation from Site 2 in absence of second Cu(I), Cu(I) dissociation from Site 2 in presence of Site 1 Cu(I), O- to IP-conformational flipping from classical MTD; and Movies: M1 – Site 1 to Site 2 Cu(I) translocation – and M2 – Site 2 Cu(I) dissociation. http://doi.org/10.1017/qrd.2022.2.

**Data availability statement.** Trajectory data for all simulations can be obtained from authors upon request.

**Author contributions.** A.M. designed and supervised the study. P.J. performed the simulations. P.J. and J.A. analysed the results. All authors wrote the manuscript.

**Financial support.** P.J. was supported by the HPC Europa Fellowship (HPC17M882U) under the Project HPC-EUROPA3 (INFRAIA-2016-1-730897). A.M. thanks the financial support of Italian Association for cancer research (AIRC) project Investigator Grant No. 24514 and the 'Against bRain cancEr: finding personalized therapies with in Silico and *in vitro* strategies' (ARES) CUP: D93D19000020007 POR FESR 2014 2020 – 1.3.b – Friuli Venezia Giulia for financial support. J.A. was supported by the 'Giovanni Fraviga' AIRC-FIRC fellowship. The authors thank Exact Lab for the computational resources available through the ARES project.

**Conflicts of interest.** The authors declare no conflicts of interest.

**Open Peer Review.** To view the open peer review materials for this article, please visit http://doi.org/10.1017/qrd.2022.2.

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
