## [Reviewer Report]

*Comments to Author*: Authors in the present manuscript perform all-atoms simulations by applying a well consolidated procedure and usual in the Magistrato lab aimed to provide insights on the protective mechanism of copper transporter 1 (CTR1) from excessively high in-cell Cu(I) levels.

In general, the manuscript is clear and well written resulting in agreement with the proposals of the authors. But before its publication minor issues have to be addressed:

How the interaction between copper and sulphur atom have been described in the classical MD? The fact that Cu FF parameters arise from MD simulations of Cys coordinating Cu-proteins, could induce some changes since the sulphur coordinating copper in Met is bound to two carbon atoms? The charge values can suffer this aspect.

- As authors can better rationalize what observed in the case of coordination geometry of copper i.e. "…CTR1 may simultaneously bind two Cu(I) ions in the selectivity filter, even though the resulting coordination geometry is slightly different from that captured in the crystal structure" (Figure S3)?

- Since many important figures, Figure S6 and S7actually in Supporting Information, are analyzed and commented in main manuscript, authors may include them in the main?

In Figure S1 the plot related to Cu-S Distance for Site 1 occupied, Site 1 Met-triad seems not be complete since it finishes at about 9 ps with respect to total of 10 ps. Please authors can check.

Some typos are present so authors are invited to revise the text.

---

## [Reviewer Report]

*Comments to Author*: Reviewer #1: Authors in the present manuscript perform all-atoms simulations by applying a well consolidated procedure and usual in the Magistrato lab aimed to provide insights on the protective mechanism of copper transporter 1 (CTR1) from excessively high in-cell Cu(I) levels.

In general, the manuscript is clear and well written resulting in agreement with the proposals of the authors. But before its publication minor issues have to be addressed:

How the interaction between copper and sulphur atom have been described in the classical MD? The fact that Cu FF parameters arise from MD simulations of Cys coordinating Cu-proteins, could induce some changes since the sulphur coordinating copper in Met is bound to two carbon atoms? The charge values can suffer this aspect.

- As authors can better rationalize what observed in the case of coordination geometry of copper i.e. "…CTR1 may simultaneously bind two Cu(I) ions in the selectivity filter, even though the resulting coordination geometry is slightly different from that captured in the crystal structure" (Figure S3)?

- Since many important figures, Figure S6 and S7actually in Supporting Information, are analyzed and commented in main manuscript, authors may include them in the main?

In Figure S1 the plot related to Cu-S Distance for Site 1 occupied, Site 1 Met-triad seems not be complete since it finishes at about 9 ps with respect to total of 10 ps. Please authors can check.

Some typos are present so authors are invited to revise the text.